evolution, ecology

Lake Tanganyika, mouthbrooding, trophic morphology, gill rakers

**Authors for correspondence:**
Fabrizia Ronco
e-mail: fabrizia.ronco@unibas.ch
Marius Roesti
e-mail: marius.roesti@iee.unibe.ch
Walter Salzburger
e-mail: walter.salzburger@unibas.ch

All authors contributed equally to this study (shared authorship).

# A functional trade-off between trophic adaptation and parental care predicts sexual dimorphism in cichlid fish

Fabrizia Ronco[1], Marius Roesti[1,2,3] and Walter Salzburger[1]

[1]Zoological Institute, University of Basel, Vesalgasse 1, 4051 Basel, Switzerland
[2]Department of Zoology, University of British Columbia, 6270 University Boulevard, Vancouver, British Columbia, Canada V6T1Z4
[3]Institute of Ecology and Evolution, University of Bern, Bern 3012, Switzerland

MR, 0000-0002-7408-4804; WS, 0000-0002-9988-1674

Although sexual dimorphism is widespread in nature, its evolutionary causes often remain elusive. Here we report a case where a sex-specific conflicting functional demand related to parental care, but not to sexual selection, explains sexual dimorphism in a primarily trophic structure, the gill rakers of cichlid fishes. More specifically, we examined gill raker length in a representative set of cichlid fish species from Lake Tanganyika featuring three different parental care strategies: (i) uni-parental mouthbrooding, whereby only one parental sex incubates the eggs in the buccal cavity; (ii) bi-parental mouthbrooding, whereby both parents participate in mouthbrooding; and (iii) nest guarding without any mouthbrooding involved. As predicted from these different parental care strategies, we find sexual dimorphism in gill raker length to be present only in uni-parental mouthbrooders, but not in bi-parental mouthbrooders nor in nest guarders. Moreover, variation in the extent of sexual dimorphism among uni-parental mouthbrooders appears to be related to trophic ecology. Overall, we present a previously unrecognized scenario for the evolution of sexual dimorphism that is not related to sexual selection or initial niche divergence between sexes. Instead, sexual dimorphism in gill raker length in uni-parental mouthbrooding cichlid fish appears to be the consequence of a sex-specific functional trade-off between a trophic function present in both sexes and a reproductive function present only in the brooding sex.

## 1. Introduction

Sexual dimorphism—that is, the different appearance of males and females within a species—is a prevalent phenomenon in animals [1,2]. However, the evolutionary processes leading to sexual dimorphism remain poorly understood in many instances [1,3]. Traits that differ between the sexes of a species can, in principle, be categorized into primary, secondary and ecological sex traits [4,5]. Primary sex traits are required functionally for reproduction and relate to organs that are specific to one sex (gonads and copulatory organs). By contrast, secondary and ecological sex traits have no direct function in reproduction and often involve modifications of characters that are shared between sexes, yet are selected towards divergent optima, thus resulting in an inter-sexual conflict [3]. Dimorphism in secondary sex traits is typically driven by sexual selection [4,5], as is the case for ornaments involved in inter-sexual selection (mate choice) or weaponry used in intra-sexual combats (mate competition) [5]. Ecological sex traits, on the other hand, are characteristics that differ between males and females as a consequence of initial ecological niche divergence between the sexes, but not due to sexual selection.

From a theoretical point of view, several models have been developed to explain purely ecology-caused sexual dimorphism [6]. Yet empirical evidence

for ecological sex traits remains scarce [7,8]. A major difficulty is to distinguish between cause and consequence, that is, whether sexual dimorphism is indeed primarily ecologically caused, or whether niche divergence between males and females is the consequence of an initially non-ecological sexual dimorphism [1]. In the latter case, sexual dimorphism in an ecological trait can be the consequence of selective forces that are not primarily related to sexual or ecological selection and that are therefore not covered by available theoretical models [1]. For example, a structure involved in food uptake and/or processing (i.e. a trophic trait) of a species could have an additional function in a reproductive behaviour without sexual selection acting on the focal trait, such as in nest-building or defending offspring [1]. A trait with such a dual function—each of which is likely to have a distinct trait optimum (a trophic and a reproductive one)—is expected to experience a trade-off (figure 1). The realized trait values should thus lie somewhere in-between the two optima (figure 1b). If the presence of a conflicting function in such a trait is restricted to only one of the two sexes, the resulting trade-off will be sex-specific too, potentially leading to sexual dimorphism (figure 1c). In such a case, the realized trait values are expected to be near the trophic optimum in one sex, while they should be shifted away from the trophic optimum towards the optimum of the conflicting (reproductive) function in the sex experiencing the trade-off. This shift in trophic morphology may subsequently result in divergent niche use between the sexes.

The gill rakers of cichlid fishes from East African Lake Tanganyika provide a rare opportunity to test, in a comparative framework, for a sex-specific trade-off related to brood care—but not to sexual selection—in an otherwise trophic trait. This is because of the important role of gill rakers (i.e. spine-like, bony protrusions of the branchial gill arches in fishes) in food uptake and handling of particles within the buccal cavity [9], the potential involvement of gill rakers in brood care in many cichlids and the different brood care strategies found among the closely related cichlids from Lake Tanganyika. More specifically, one particular feature of gill rakers, gill raker length, has been shown to be strongly associated with trophic ecology in many fish [10–14], including cichlids [15,16], a pattern we here corroborate for gill raker length across 65 Tanganyikan cichlid species (figure 2a).

All Tanganyikan cichlids provide intensive parental brood care, either in the form of bi-parental mouthbrooding (both sexes participate in parental care), uni-parental mouthbrooding (only one sex—in the case of Tanganyikan cichlids the female—participates in parental care) or substrate spawning with nest guarding (parental care does not involve any form of mouthbrooding) [17]. Mouthbrooding species incubate their brood in the buccal cavity until the eggs' yolk sac is used up and the fry becomes free-swimming. During this entire period, which in Tanganyikan cichlids lasts between 6 and 30 days, the fertilized eggs—and later also the growing larvae—are in close physical contact with the gill rakers (figure 2b) and are regularly 'churned' inside the buccal cavity, probably to facilitate their ventilation and cleaning [18,19]. Gill rakers in mouthbrooding cichlids are thus expected to not only function in the uptake and handling of food particles, but also in the retention and handling of the eggs and larvae in the buccal cavity. Indeed, changes in head morphology have previously been associated with mouthbrooding [16–20], and sexual dimorphism in gill raker

length has been reported for *Astatotilapia burtoni*, a uni-parental mouthbrooding cichlid from the Lake Tanganyika basin [15]. Taken together, mouthbrooding emerges as a promising candidate for an additional and probably conflicting functional demand of gill rakers.

In this study, we hypothesized that breeding mode can predict sexual dimorphism in gill raker length in Lake Tanganyika cichlids, whereby the three different breeding modes exemplify the three scenarios illustrated in figure 1. (i) In non-mouthbrooders, gill rakers are expected to have evolved relatively unconstrained towards the trophic trait optimum in both sexes (figure 1a). (ii) In bi-parental mouthbrooders, gill raker morphology should be influenced by both feeding and parental care (mouthbrooding). These two functions are unlikely to have identical trait optima, but the optimum resulting from the trade-off should be the same for both sexes (figure 1b). (iii) In uni-parental mouthbrooders, the functional trade-off between feeding and parental care should only occur in the mouthbrooding sex (females), whereas gill raker morphology in the non-mouthbrooding sex (males) should be selected towards the trophic optimum (figure 1c). Sexual dimorphism in gill raker length should thus occur exclusively in uni-parental mouthbrooders, but not in bi-parental mouthbrooders nor in non-mouthbrooding substrate brooders. The direction of the sexual dimorphism in uni-parental mouthbrooders is, however, hardly predictable as it should depend on the relative position of the two conflicting trait optima with respect to each other, which may well be species-specific. Finally, we hypothesized that trophic ecology determines the strength of the conflict (i.e. how divergent the two conflicting optima are) as a result of different trait optima in different trophic niches. To test these hypotheses, we examined a representative set of cichlid species for sexual dimorphism in gill raker length and tested for an association with breeding mode and trophic ecology.

## 2. Material and methods

### (a) Sampling
Samples were collected between 2014 and 2017 during several field trips to the southern part of Lake Tanganyika, under the research permits number 005937 (F.R.) and 004273 (W.S.) issued by the Republic of Zambia. Combined with available data on gill raker length from additional Tanganyikan cichlid species [16], the final dataset covered 65 species, well representing the phylogenetic (13 out of 16 tribes [21]), eco-morphological and behavioural (breeding modes) diversity of the species-flock of cichlid fishes in Lake Tanganyika (see electronic supplementary material for detailed information on the sampling procedure and electronic supplementary material, table S1 for sample sizes).

### (b) Stable isotopes
We assessed the trophic ecology of all species by quantifying stable isotope signatures of carbon (C) and nitrogen (N) in typically 10 specimens per species ($n = 661$). The ratios between the rare isotopes $^{13}C$ to $^{12}C$ ($\delta^{13}C$) and $^{15}N$ to $^{14}N$ ($\delta^{15}N$) inform about two major components of aquatic ecology, the benthic–pelagic ($\delta^{13}C$) and trophic ($\delta^{15}N$) position within an ecosystem [22]. This method has previously been applied to Tanganyikan cichlids and was compared to stomach content data [21], permitting an interpretation of food types. In this study, we analysed dried muscle tissue (from the epaxialis between the head and the dorsal fin) with a Flash 2000 elemental analyser coupled to

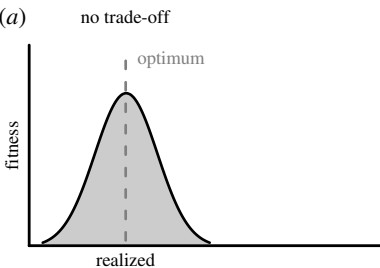
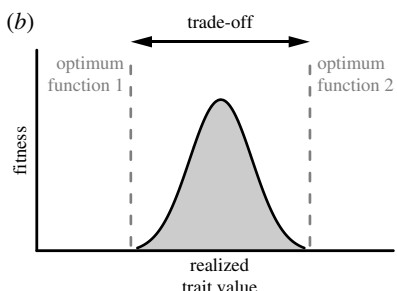
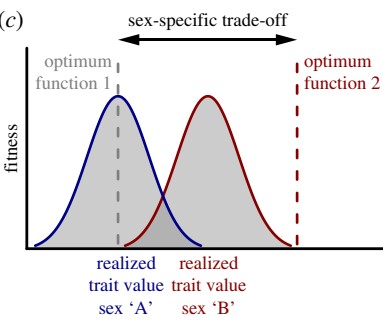

**Figure 1.** Proposed scenario for how two conflicting functions of the same trait can result in sexual dimorphism. (*a*) No trade-off, single function: the trait value is selected towards a single functional optimum resulting in an overlap between the optimal and the realized trait value. (*b*) Trade-off between two conflicting functions of the trait (two divergent functional optima): selection is likely to favour an intermediate phenotype (solid line), deviating from both functional trait optima (dashed lines). (*c*) Sex-specific trade-off between two conflicting functions, with a single functional optimum for one sex (sex 'A') and two conflicting optima for the other sex (sex 'B'): different selective outcomes are expected. In sex 'A', the trait is selected towards the functional optimum '1'. Hence, the realized trait value for sex 'A' (blue line) is likely to overlap with the optimum (although genetic constrains could lead to a deviation; not shown). In sex 'B', however, the trade-off between the two conflicting functional optima (dashed lines) is likely to result in intermediate realized trait values (red line). (Online version in colour.)

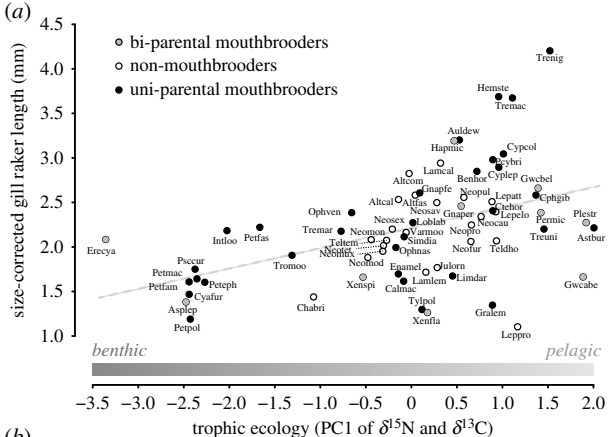

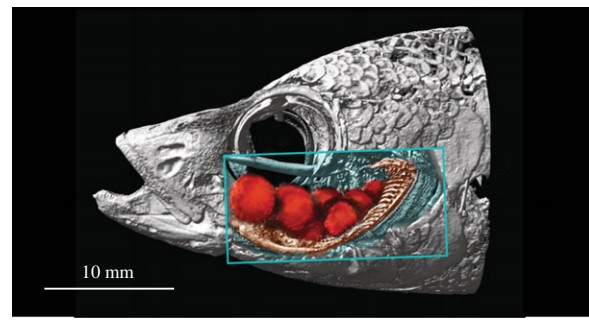

**Figure 2.** Gill rakers in Lake Tanganyika cichlids and their association with trophic ecology. (*a*) Phenotype–environment correlation between size-corrected gill raker length and trophic ecology (PC1 scores of stable isotope data). Longer gill rakers are associated with pelagic feeding, and shorter gill rakers with benthic feeding. This benthic–pelagic feeding trajectory is indicated above the *x*-axis (see electronic supplementary material, figure S1*a*). Data points represent species means and are shaded according to the breeding mode of the species (see electronic supplementary material, table S1 for full species names). (*b*) CT scan of a mouthbrooding *Paracyprichromis* sp. female (see electronic supplementary material for details on scanning and processing). Parts of the skull were virtually removed (box), revealing the developing eggs in the buccal cavity (highlighted in red) and the gill raker apparatus (highlighted in brown). (Online version in colour.)

a Delta V Plus continuous-flow isotope ratio mass spectrometer (IRMS) via a Conflo IV interface (Thermo Fisher Scientific, Bremen, Germany) in the SLU-Lab at the University of Basel (data have been deposited on the Dryad Digital Repository: https://doi.org/10.5061/dryad.fm4707v [23]). We then used a

principal component analysis (PCA) to integrate over the $\delta^{15}$N and $\delta^{13}$C stable isotope ratios to obtain a univariate metric for trophic ecology.

## (c) Gill raker morphology

We measured gill raker length under a binocular (Leica MZ7$_5$) as described previously [10,16]. Measurements were taken by two investigators on blinded samples (F.R., M.R.) and recorded by a third investigator (W.S.). Measurements were converted to millimetre scale and averaged across the three gill rakers measured per specimen (i.e. the second, third and fourth raker on the first branchial gill arch). To avoid a potential investigator bias, samples were assigned randomly to one of the two investigators. We measured gill raker length in 508 specimens (38 species). In combination with data from Muschick *et al.* [16], we obtained a dataset comprising 935 specimens and 65 species (data have been deposited on the Dryad Digital Repository: https://doi.org/10.5061/dryad.fm4707v [23]). Gill raker length was strongly correlated with body size (standard length = SL; Pearson's $r = 0.68$, $p < 0.001$), and thus, size corrected prior to further analysis. Size correction was done specifically for each analysis (see below).

## (d) Phenotype–environment correlation

To investigate how gill raker length is associated with trophic ecology, we size-corrected gill raker length of each specimen using residuals from a common linear model applied across all specimens from all 65 species (with gill raker length as response variable and SL as explanatory variable; $R^2 = 0.46$, $p < 0.001$). We then added the value of the largest residual to restore positive values in the initial measuring unit (mm). The species mean of these size-independent values and the PC1 scores of stable isotope data were used to test for a phenotype–environment correlation using a linear model and Pearson's $r$ statistics. Statistical significance was assessed using 10 000 random permutations of the observed species means over the stable isotope PC1 scores [24]. All $p$-values and 95% confidence intervals in this paper were obtained through analogous resampling procedures, except for analyses accounting for phylogenetic relationships. To account for phylogenetic dependence of the species, we applied a 'phylogenetic generalized least squares' fit using the R package *caper* [25]. For all analyses incorporating phylogenetic relationships, we used the phylogenetic hypothesis from Colombo *et al.* [26] and pruned it to the set of taxa present in our datasets. One species (*Petrochromis ephippium*) was not represented in the phylogenetic tree and was therefore omitted from these analyses.

## (e) Sexual dimorphism

To test for sexual dimorphism in gill raker length in non-, bi- and uni-parental mouthbrooders, we focused on a subset of species ($n = 20$) for which sex information was available. Here, size correction of gill raker length was performed separately for each species, using species-specific linear models, maximizing comparability between the sexes. We then tested for a difference in the length of male and female gill rakers within each species, and whether the grand-mean per breeding mode deviated from zero. We further evaluated whether the *extent* of the dimorphism irrespective of directionality (i.e. the *absolute* difference of female minus male gill raker length per species) differed among the breeding modes by calculating *F*-statistics across the three groups (ANOVA), followed by pairwise comparisons of the breeding modes. To account for phylogenetic dependence of the species, we applied a phylogenetic ANOVA using the function phylANOVA from the R package *phytools* [27].

Finally, we tested for an association between the extent of sexual dimorphism and trophic ecology (PC1 scores of stable isotope data) within uni-parental mouthbrooders. Based on a Davies test [28], which tests for a breakpoint in a linear relationship between two variables, we fitted a segmented regression model [28]. Note that the reported *p*-values for the Davies test were not obtained through permutation, but were taken directly from the output of the davies.test function as implemented in the R package *segmented* [28]. To validate the results in a phylogenetic framework, we used the estimated breakpoint in PC1 scores from the segmented regression model as a threshold to assign the uni-parental mouthbrooders into two trophic groups and tested for a difference in the extent of sexual dimorphism between these groups using a phylogenetic ANOVA. All graphing and statistical analyses were conducted in R [29].

## 3. Results

### (a) Gill raker length is associated with trophic ecology

A PCA of the stable isotope ratios of nitrogen ($\delta^{15}$N) and carbon ($\delta^{13}$C) was used to reduce dimensionality of the two components of trophic ecology. This allowed working with a univariate proxy for trophic ecology. PC1 explained 77.3% of the total variation in the stable isotope data, and was loaded negatively for $\delta^{13}$C (−0.71) and positively for $\delta^{15}$N (0.71) (electronic supplementary material, figure S1*a*). Higher PC1 scores thus reflected pelagic feeding (e.g. on zooplankton and/or fish fry) and a relatively high position in the food chain (hereafter simply referred to as 'pelagic'), whereas benthic/littoral species with a mainly algivorous feeding lifestyle and a lower trophic position had lower PC1 scores (hereafter simply called 'benthic'). Gill raker length was positively associated with trophic ecology across the 65 species (Pearson's $r = 0.46$, $p < 0.001$; $R^2 = 0.20$, $p < 0.001$), with shorter gill rakers in benthic and longer gill rakers in pelagic species (figure 2*a*; electronic supplementary material, figure S1*b*). This result held true after accounting for phylogenetic dependence of the trait values ($R^2 = 0.14$, $p = 0.002$, $\lambda = 0.43$).

### (b) Sexual dimorphism is predicted by breeding mode and trophic ecology

Sexual dimorphism in size-corrected gill raker length was pronounced in uni-parental mouthbrooders, and reached statistical significance ($p < 0.05$) in three out of nine species (see electronic supplementary material, table S2*a*). By contrast, none of the bi-parental mouthbrooding species, nor

any substrate brooding species, showed evidence for sexual dimorphism (figure 3*a*).

The grand mean per breeding mode of the difference between male and female gill raker length did not deviate from zero in any of the three breeding modes (see electronic supplementary material, table S2*b*). However, uni-parental species showed a strongly increased variation in sexual dimorphism compared to bi-parental mouthbrooders and non-mouthbrooders (figure 3*a*). The *absolute* difference in gill raker length between the sexes revealed a significantly greater extent of sexual dimorphism in uni-parental mouthbrooders compared to bi-parental and non-mouthbrooding species in an ordinary ANOVA ($F = 6.19$, $p = 0.007$) (figure 3*b*; electronic supplementary material, table S2*c,d*). When accounting for phylogenetic dependence, only uni-parental and bi-parental mouthbrooders showed a difference in the extent of sexual dimorphism ($p = 0.022$).

Finally, we focused on the association between the extent of sexual dimorphism and trophic ecology within uni-parental mouthbrooders. We found a statistically supported breakpoint in the linear relationship between sexual dimorphism and trophic ecology ($p = 0.04$). The fitted segmented model estimated a breakpoint at a PC1 score of 0.34, with PC1 scores higher than 0.34 showing a strong positive association with the extent of sexual dimorphism (figure 4; electronic supplementary material, table S3). When using this estimated breakpoint to assign the species into two trophic groups and accounting for phylogenetic dependence, the species with higher PC1 scores showed a distinctly greater extent of sexual dimorphism than the species with PC1 scores below the threshold ($F = 22.8$, $p = 0.004$).

## 4. Discussion

In this study, we addressed the question of whether a conflicting (sex-specific) functional demand linked to parental care can explain sexual dimorphism in an otherwise trophic trait. To this end, we investigated gill raker length in a set of cichlid fish species from Lake Tanganyika covering three different breeding modes and a variety of trophic ecologies (figure 2*a*).

Gill rakers are an important structure for uptake and handling of food in the buccal cavity in fish [9], and the length of gill rakers is generally associated with different trophic ecologies: pelagic species feeding on small and mobile prey commonly have longer gill rakers, while benthic species feeding on larger and immobile prey (or *aufwuchs*) have shorter gill rakers [10–16]. Here we corroborate this phenotype–environment correlation in an extensive dataset covering 65 cichlid species from Lake Tanganyika, representing the morphological, ecological and phylogenetic diversity of the lake's cichlid assemblage: we find an association between gill raker length and trophic ecology (as approximated by the PC1 of stable isotope data), with longer gill rakers in cichlids with more pelagic stable isotope signatures, and shorter gill rakers in species with more benthic signatures (figure 2*a*). Based on a previous study linking stable isotope signatures with stomach content analysis in Tanganyika cichlids [21], we conclude that pelagic stable isotope signatures usually correspond to invertebrate/zooplankton/small fish feeders, whereas species with benthic signatures predominantly feed on algae and plants. Note, however, that also predatory species feeding on large fish show pelagic

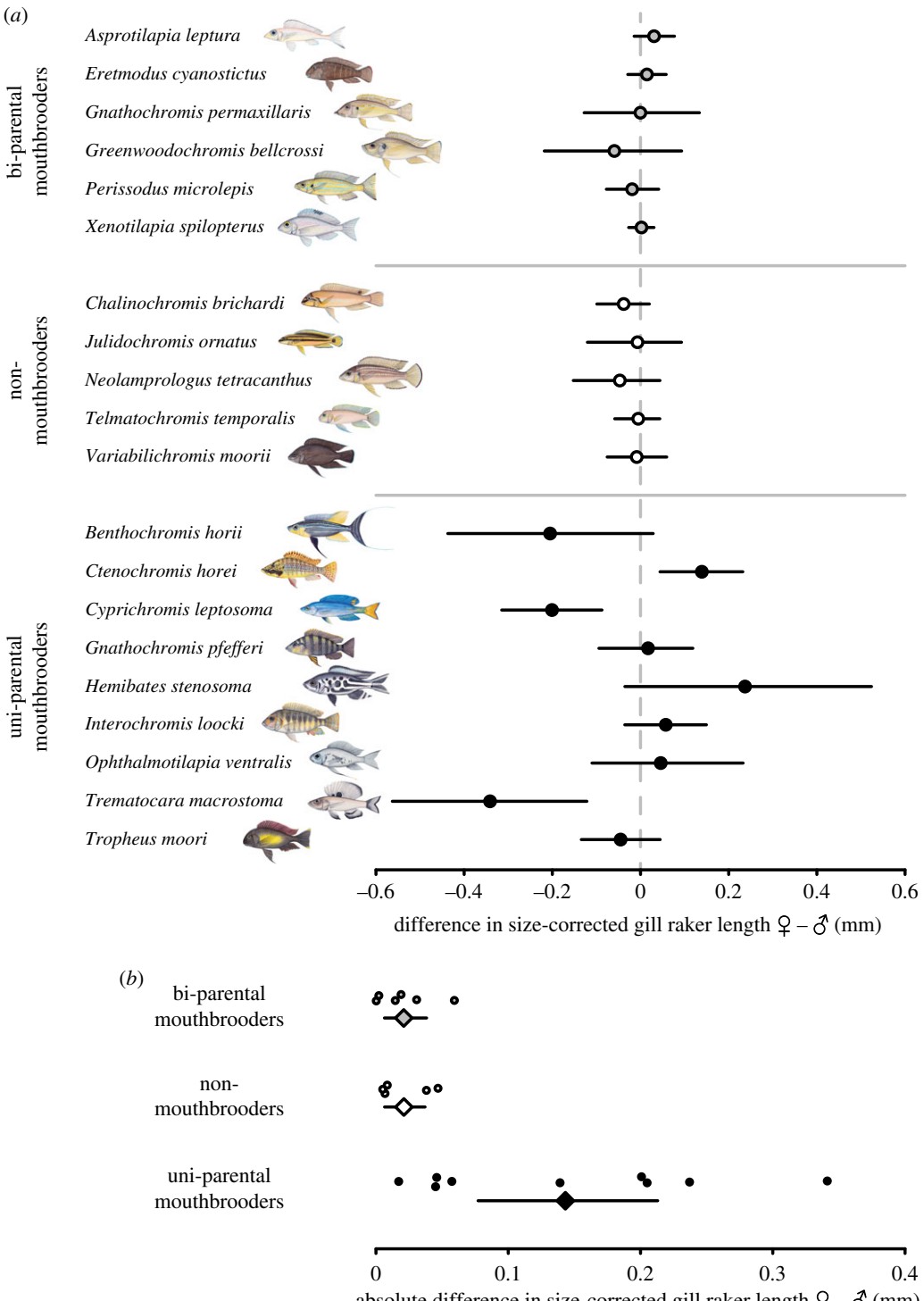

**Figure 3.** Sexual dimorphism in gill raker length. (a) Female minus male size-corrected gill raker length per species shows pronounced sexual dimorphism in uni-parental mouthbrooders compared to bi-parental and non-mouthbrooding species. (b) Extent of sexual dimorphism, calculated by the *absolute* difference of female minus male size-corrected gill raker length (circles are species means and squares are grand-means per breeding mode). Uni-parental mouthbrooders show an increased extent of sexual dimorphism compared to bi-parental and non-mouthbrooding species (nest guarders). All error bars represent 95% confidence intervals of the means; see electronic supplementary material, tables S1 and S2 for sample sizes and p-values. (Online version in colour.)

signatures, but have rather short gill rakers (see e.g. *Lepidio-lamprologus profondicula*; 'Leppro'; figure 2a).

We hypothesized that gill raker length is also relevant for mouthbrooding, thus resulting in a conflicting functional demand of gill raker morphology in addition to food update and handling (figure 1b). Mouthbrooding is a particular form of parental care and widespread among cichlid fishes, where it occurs in a uni-parental (maternal or paternal) or bi-parental mode. Mouthbrooding is a costly

trait [30] and has been reported to induce morphological changes including an enlargement of the head or the buccal cavity [20,31–33], or a reduction in gill size [34]. Gill raker length has, however, not yet been examined in the context of mouthbrooding. This is surprising given that gill rakers are expected to be functionally involved in mouthbrooding, either directly via the active handling of the eggs or larvae [18], or indirectly through the close physical contact between gill rakers and the offspring (figure 2b).

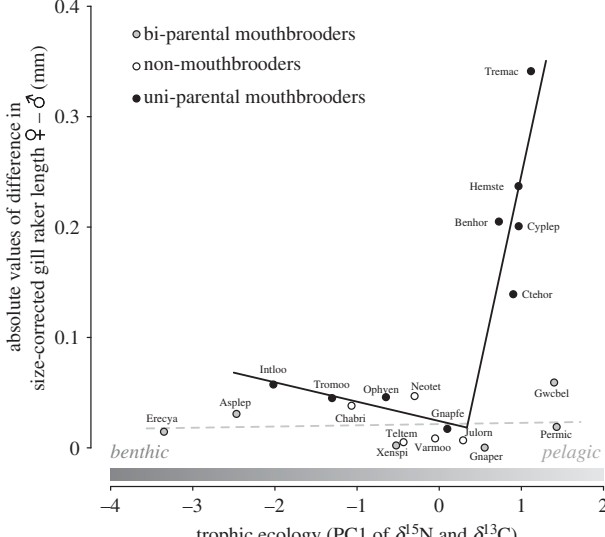

**Figure 4.** Association between the extent of sexual dimorphism and trophic ecology: in uni-parental mouthbrooders, the linear association between the extent of sexual dimorphism (i.e. absolute difference in size-corrected gill raker length between females and males) and trophic ecology shows a break-point at a trophic PC1 score of 0.34. For species above this threshold, PC1 scores and sexual dimorphism correlate strongly and positively. The solid line represents the result of a segmented regression model fitted for uni-parental mouthbrooders. As no differences were found between bi-parental mouthbrooders and non-mouthbrooders, the two groups were pooled in a linear model (dashed grey line). Note that the five uni-parental mouthbrooding species with the largest sexual dimorphism belong to five different tribes (see electronic supplementary material, figure S1*c*). The benthic–pelagic feeding trajectory represented by PC1 scores is indicated above the *x*-axis (see electronic supplementary material, figure S1*a*).

Under the assumption that gill rakers experience different selective regimes among the three breeding modes (bi-parental mouthbrooding, uni-parental mouthbrooding and non-mouthbrooding) due to their dual function in both feeding and breeding, we predicted sexual dimorphism to be present exclusively in uni-parental mouthbrooders (figure 1). We examined 20 Tanganyikan cichlid species and investigated how well breeding mode and/or feeding ecology can explain sexual dimorphism. Indeed, we found males and females of non- and bi-parental mouthbrooding species not to differ in gill raker length. By contrast, several of the uni-parental mouthbrooding species were sexually dimorphic in gill raker length, varying in extent and direction (figure 3*a*). The overall difference between non- and bi-parental mouthbrooders relative to uni-parental mouthbrooders became particularly evident when comparing the *absolute* extent of sexual dimorphism (i.e. sexual dimorphism irrespective of its direction) among breeding modes (figure 3*b*). When accounting for phylogeny, the difference between uni-parental and bi-parental mouthbrooders in the extent of sexual dimorphism was confirmed, but not so for non-mouthbrooders. This is hardly surprising, given the monophyly of the vast majority of non-mouthbrooding cichlids in Lake Tanganyika (the Lamprologini, which make up 50% of all species in that lake; see electronic supplementary material, figure S1*c*), thus reducing statistical power in comparisons involving non-mouthbrooders. Nevertheless, the difference in the extent of sexual dimorphism between uni-parental and bi-parental mouthbrooders supported the idea that breeding

mode can partially predict the presence or absence of sexual dimorphism in gill raker length.

How can the variation in the extent and direction of sexual dimorphism *within* uni-parental mouthbrooders be explained? Under the assumption of a trade-off between a trophic versus reproductive function of gill rakers, both the extent and the directionality of the sexual dimorphism should depend on the relative position of the two optima in relation to one another (figure 1). Clearly, the association of gill raker length and trophic ecology across the 65 cichlid species suggests species-specific optima for gill raker length (figure 2*a*). Although the factors determining optimal gill raker length for mouthbrooding are unknown, life-history traits such as clutch and egg size or breeding duration are likely to be relevant. Unfortunately, data on life-history traits are too scarce (and/or too vague) to allow testing for an association with gill raker length. Nevertheless, clutch size emerges as a promising candidate trait to explain variation in the direction of sexual dimorphism among uni-parental mouthbrooders (see electronic supplementary material, figure S2*a*). On the other hand, since life-history traits differ among cichlid species [35], the reproductive optimum of gill raker length is expected to be species-specific too. This is further corroborated by the difference in the *directionality* of sexual dimorphism in gill raker length among uni-parental mouthbrooders with respect to actual gill raker length (electronic supplementary material, figure S2*b*). Thus, the finding that some uni-parental mouthbrooders show a female-biased dimorphism (longer gill rakers in females), while others show a male-biased dimorphism, is likely to reflect variation in the relative position of the conflicting trait optima.

Likewise, the absence of any sexual dimorphism in some of the uni-parental mouthbrooders might be the result of overlapping trait optima for the two functional demands. Species with extreme trophic ecologies may be expected to generally experience stronger deviations between the trophic and reproductive optima than species with intermediate trophic ecologies. Additionally, variation in the extent of sexual dimorphism among uni-parental mouthbrooders might be the result of similarly strong selection towards the optimum for mouthbrooding in all species, but varying selection regimes with respect to the optimal trait value for feeding, depending on the trophic ecology of a species. For example, if gill raker morphology is of particular importance for efficient food uptake and handling in a species (as in pelagic suction feeders [11]), the selective pressures acting antagonistically are expected to be strong and a dimorphism is more likely to be expressed. On the other hand, in species where the gill rakers are less important for feeding (as in benthic algivores), sexually antagonistic selection would be unbalanced, resulting in a less pronounced or no sexual dimorphism. Accordingly, in both cases, the differences in the extent of sexual dimorphism in uni-parental mouthbrooders are expected to depend on the trophic ecology of the species. We tested this prediction and found uni-parental mouthbrooding species to show an association between the (absolute) extent of sexual dimorphism and trophic ecology. This association was not linear along the entire trophic continuum, but rather increased rapidly after a certain breakpoint (figure 4). This implies that whether or not a sexual dimorphism in gill raker length occurs depends on both the breeding mode and the trophic ecology of a species, with breeding

mode determining the potential for a sex-specific functional conflict, and trophic ecology determining the strength of the conflict.

With the data at hand, we cannot formally test for the strength of selection acting on gill raker length, nor can we directly measure the optima in trait value for feeding versus mouthbrooding. Hence, we cannot disentangle cases where the strength of the conflict depends on how balanced the selective pressures are that act on the two optima, on how divergent the two optima are, nor a combination of both. Nevertheless, our findings provide empirical evidence for the scenario that a sex-specific functional conflict due to parental care by only one sex of a species explains sexual dimorphism in a trait.

The finding of sexual dimorphism to be present exclusively in uni-parental mouthbrooders largely contradicts predictions from popular models of ecology-caused sexual dimorphism: if inter-sexual competition for resources were the trigger for sexual dimorphism [6], one would expect sexual dimorphism to occur mainly in species forming pair bonds and sharing feeding and breeding territories [36]. In our study system, this applies primarily to bi-parental mouthbrooders and non-mouthbrooders (bi-parental nest guarders), but not to uni-parental mouthbrooders. Other ecological models for sexual dimorphism, such as the 'bimodal niche model' [6] (two alternative optima in trait value exist, followed by disruptive selection between the sexes) or the 'dimorphic niche model' [6] (intrinsic differences between males and females in energetic needs lead to niche divergence between the sexes), would also not predict sexual dimorphism to occur exclusively in uni-parental mouthbrooders. Moreover, most models of ecology-caused sexual dimorphism assume niche divergence between the sexes. However, such a difference in niche use between males and females is not evident from our stable isotope data (electronic supplementary material, figure S3).

Unlike most studies investigating causes of sexual dimorphism in relation to ecology [7,8,37,38], we can largely exclude the possibility that sexual selection has directly driven or reinforced the observed sexual dimorphism. This is because gill rakers are cryptic to the outer appearance of a fish and thus highly unlikely to serve as a signal in mate choice or mate competition. One could of course argue that sexual selection initially contributed to the evolution of sex-specific roles in breeding behaviour, but here we refer to sexual selection acting directly on the focal trait. Taken together, sexual dimorphism in our study system is unlikely to be explained by sexual selection or initial niche divergence between the sexes, thus providing a novel view on the evolution of sexual dimorphism in nature.

Although our study provides an explanation why gill raker length differs between the sexes in some cichlid species, but not in others, it remains an open question how sexual dimorphism in this trait is achieved developmentally. Variation in gill raker length has been shown to have a largely genetic basis in threespine stickleback [39,40], and a common garden experiment with divergent A. burtoni cichlid ecotypes revealed both a genetic and a plastic component in gill raker length variation [15]. What remains to be tested is the degree to which sexual dimorphism in gill raker length of cichlids is genetically based or is the result of a plastic response to mouthbrooding. It would further be interesting to investigate other components of the cichlids' trophic morphology with respect to sexual dimorphism and parental care.

In conclusion, our study establishes an overall phenotype–environment association between gill raker length and trophic ecology across 65 Tanganyikan cichlid species, and reveals that gill raker morphology is influenced by mouthbrooding. As a consequence, the presence and extent of sexual dimorphism in gill raker length is predicted by both the breeding mode and the trophic ecology of a species. Sexual dimorphism in gill raker length of uni-parental mouthbrooding cichlids is unlikely to be explained by sexual selection or initial niche divergence between the sexes, but instead is caused by a sex-specific trade-off between two conflicting functional demands of the same trait, one related to trophic adaptation and one to parental care.

Data accessibility. Data available from the Dryad Digital Repository: https://doi.org/10.5061/dryad.fm4707v [23].

Authors' contributions. M.R. conceived the study, with all authors making later contributions to the study design. F.R. and W.S. collected the samples new to this study in the field. All authors contributed to measuring gill rakers. F.R. obtained stable isotope data, analysed and visualized the data with input from M.R. and W.S. F.R. drafted the manuscript, with all authors contributing to the writing of the final paper.

Competing interests. We declare we have no competing interests.

Funding. This study was funded by European Research Council (ERC, Consolidator Grant 'CICHLID~X') to W.S., and grants from the Swiss National Science Foundation to M.R. (P2BSP3_161931, 300PA_174344) and W.S. (156405).

Acknowledgements. We would like to thank Adrian Indermaur, Athimed El Taher, Ann-Christin Honnen, Lukas Widmer, Elia Heule and Anna Boila for their help in the field, and Ansgar Kahmen, Victor Evrard and Anna Boila for their assistance with stable isotope data collection. We further thank Daniel Berner for statistical advice, Hanna Kokko and Hugo Gante for valuable discussions, Lukas Schärer for helpful comments on the manuscript, Julie Himes for fish illustrations in figure 3a, as well as Peter Wainwright and three anonymous reviewers for helpful comments that improved this manuscript.

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
