## [Reviewer comments · Proceedings of the Royal Society B: Biological Sciences]

Review History

RSPB-2019-0023.R0 (Original submission)

Review form: Reviewer 1

Recommendation

Reject – article is scientifically unsound

Scientific importance: Is the manuscript an original and important contribution to its field?

Marginal

General interest: Is the paper of sufficient general interest?

Marginal

Quality of the paper: Is the overall quality of the paper suitable?

Acceptable

Is the length of the paper justified?

Yes

Should the paper be seen by a specialist statistical reviewer?

No

Do you have any concerns about statistical analyses in this paper? If so, please specify them explicitly in your report.

Yes

It is a condition of publication that authors make their supporting data, code and materials available - either as supplementary material or hosted in an external repository. Please rate, if applicable, the supporting data on the following criteria.

Is it accessible?

Yes

Is it clear?

No

Is it adequate?

Yes

Do you have any ethical concerns with this paper?

No

Comments to the Author

The manuscript "A functional trade-off between trophic adaptation and parental care leads to sexual dimorphism" provides an interesting analysis of gill raker sexual dimorphism in cichlid fishes from Lake Tanganyika. The hypotheses set up in this work are interesting. However, I have several strong reservations about the manuscript including the framing of the paper around the idea of optima, the novelty of data and breadth of the conclusions presented, and the rigor of the evolutionary analyses of trait associations.

The discussion of optima in the introduction is interesting and makes up a substantial part of the introduction and figure 1, but little about optima is really tested in the paper. There are now phylogenetic comparative methods available that would allow the testing of trait optima associated with particular ecological phenotypes such as:

For instance:

Mahler DL, Ingram T, Revell LJ, Losos JB. 2013 Exceptional convergence on the macroevolutionary landscape in island lizard radiations. *Science* 341, 292–295.

I felt the discussion of optima and the focus of figure 1 on optima and fitness could provide a misleading view of how these traits are diverging in these cichlid fishes and what the manuscript actually tests. The focus on optima in studies of adaptation has a long history of being criticized and the current discussion without any explicit tests of "optimality" I think is unjustified.

Also, it was unclear in several cases what was novel about the data presented. Were the values of stable isotopes taken from other published papers? If so, it would be good to clarify how much of this data is unique to this paper. Also, how many of the gill raker measurements are novel to this paper? It should be made clear how many are already published in Muschick et al. 2014 and what proportion/samples are novel to this work. Since the paper seems to largely center around the reanalysis of previously published data and the conclusions are not likely to have a large impact on our general understanding of sexual dimorphism, the general reuse of data to address the narrow questions posed about sexual dimorphism will likely have limited interest to a wide audience.

Finally, and perhaps most problematically, the data is not analyzed in a rigorous evolutionary framework. For instance, in Figure 2: Are the uni-parental male trait values used since these are predicted, as the manuscript makes a case, near the optima? For a correlation like this, I would assume only the male traits should be used. Also, if this correlation uses size-corrected gill raker lengths this should be stated in the figure axes and/or legend. Most critically, this correlation should be performed in a phylogenetically explicit framework. The current correlation is statistically invalid. Additionally, the current statistical analyses of sexual dimorphism in gill raker length does not seem to adequately account for phylogeny. A phylogenetic ANCOVA should be used to show that the gill raker lengths show differences in sexual dimorphism among the three different breeding modes. With only 20 species sampled, this is unfortunately unlikely to be significant for the different parental modes. However, without adequately accounting for phylogeny it is difficult to say that the correlations reported are not spuriously due to species sampling and an inflation of presumed power due to the OTUs being compared without a phylogenetic framework.

The manuscript also has a too substantial number of grammatical, formatting, or spelling mistakes.

Here are a few:

Line 29: "dimorphism in a primarily trophic structure"

Line 153: Strange font formatting "covered 65 species, well"

Line 560: "scull" should be "skull"

Review form: Reviewer 2 (Peter Wainwright)

Recommendation

Reject – article is scientifically unsound

Scientific importance: Is the manuscript an original and important contribution to its field?

Excellent

General interest: Is the paper of sufficient general interest?

Excellent

Quality of the paper: Is the overall quality of the paper suitable?

Acceptable

Is the length of the paper justified?

Yes

Should the paper be seen by a specialist statistical reviewer?

No

Do you have any concerns about statistical analyses in this paper? If so, please specify them explicitly in your report.

Yes

It is a condition of publication that authors make their supporting data, code and materials available - either as supplementary material or hosted in an external repository. Please rate, if applicable, the supporting data on the following criteria.

Is it accessible?

Yes

Is it clear?

Yes

Is it adequate?

Yes

Do you have any ethical concerns with this paper?

No

Comments to the Author

Simply put, this is one of the most fascinating and engaging manuscripts I have ever reviewed. I don't mean that the writing is great (it's fine), I mean the story is great. What a remarkable set of insights the authors have brought here. This is just glorious stuff. The idea that the gillrakers in uniparental species differ in length as a function of where the optimal length is for the feeding ecology is just outrageously interesting in the way that it produces so much diversity in raker length.

Now, having said that I am concerned about a serious methodological issue in the work, that may substantially weaken the results. The problem is that there are no phylogenetic corrections in any of the analyses, while the study is comparative in nature. In this day and age you just can't do this kind of a study without accounting for phylogenetic relationships among the taxa under consideration. I know it means a lot more work, and I also know it means the authors could well lose some of their significant results, but this needs to be dealt with. The relationships shown in Fig 2a, 3 and 4 all require the incorporation of phylogenetic relationships into any tests.

A second, but less concerning issue is that the way that trophic ecology is quantified (the C:N PC1) ultimately is unsatisfying when interpreting what is going on with the uniparental species. I think what one wants here is a more in depth discussion of how the gill rakers are functioning in these different species and what exactly the fish are eating. Is what they are eating strongly consistent with the story that is being told here that the magnitude and direction of dimorphism depends on the optimal length for the trophic habit? Hopefully there is more specific information available about what some of these fish are actually eating - beyond the hard-to-interpret isotope data.

It would also add to the story if the authors had some sense for how the gill rakers function in fish mouthbrooding eggs and larvae. What trophic features of gill rakers are problematic for brooding rakers? Why are rakers of different length adaptive for different prey? The issue here is that there should be a performance trade-off in the ecology, with long rakers being good for some prey but also for short rakers having a performance benefit for other prey. What are these performance benefits?

L149. Should be 'field' not 'filed'.

L168. It does not appear that any attempt was made to sample rakers from a specific region of the gill arch? Are rakers the same length at different positions along the arch? In many fish they are

not. This would suggest that these relationships could be affected by where on the arch the rakers come from.

L210. Did you look to see if magnitude of dimorphism depends on body size? This could complicate analyses. I'm confused about whether this was accounted for here.

L223. Unclear what is meant by a 'scaled' PCA? I've never seen this term used before.

L276. Well, pelagic species that feed on tiny but mobile animals often have long gill rakers (as do sediment sifters that feed on tiny but mobile prey. But pelagic species that feed on large prey like fish do not have long gill rakers. None of these fish are really suspension feeders, they are suction feeders that are using gill rakers to prevent small, mobile prey from escaping from the buccal cavity into the opercular cavity after capture.

L283. I wish I could agree with this statement, but I think the authors are making assumptions about what prey underlie the isotope scores without showing the reader what they are actually eating. The manuscript shows shorter rakers in species that are low on the C:N axis and often longer rakers in fish that are high on the C:N axis (but definitely not always). But, if we don't actually know in this study what prey these fish are eating this creates some confusion.

-Peter Wainwright

Review form: Reviewer 3

Recommendation

Accept with minor revision (please list in comments)

Scientific importance: Is the manuscript an original and important contribution to its field?

Good

General interest: Is the paper of sufficient general interest?

Good

Quality of the paper: Is the overall quality of the paper suitable?

Good

Is the length of the paper justified?

Yes

Should the paper be seen by a specialist statistical reviewer?

Yes

Do you have any concerns about statistical analyses in this paper? If so, please specify them explicitly in your report.

Yes

It is a condition of publication that authors make their supporting data, code and materials available - either as supplementary material or hosted in an external repository. Please rate, if applicable, the supporting data on the following criteria.

Is it accessible?

Yes

Is it clear?

Yes

Is it adequate?

Yes

Do you have any ethical concerns with this paper?

No

Comments to the Author

RSPB-2019-0023: A functional trade-off between trophic adaptation and parental care leads to sexual dimorphism. By Ronco et al.

This is a very interesting study that investigated the relationship between sexual dimorphism in gill raker length and parental care patterns or trophic ecology. Using Tanganyika cichlid fishes with different parental care patterns and feeding ecology, they found that a functional trade-off between them, which may lead to differences in gill raker length between the sexes. The results of this study are clearly stated, which would add much contribution to our understanding of this study field. I have, however, a number of minor concerns, which were listed below.

1. Title: In general, sexual dimorphism means the dimorphism in secondary sex traits driven by sexual selection, such as coloration, weaponry and body size. However, this study deals with the sexual dimorphism in gill raker length. Hence, I suggest the authors adding some words in the title, e.g. "sexual dimorphism in gills of (cichlid) fishes".

2. Methods: The authors compared ecological traits among many cichlid species. Comparative methods, such as phylogenetic generalized least squares (PGLS), are generally used to examine the evolutionary process of these traits, but this study did not adopt such methods. Please clarify.

3. Discussion L314-330 and figure 3: The authors argued about sexual dimorphism within uni-parental mouthbrooders. I also agree that the differences in the extent of sexual dimorphism in gill rakers among uni-parental mouth brooders may be related to clutch size, egg size and the duration of parental care. If the authors have samples or data and find some papers describing these data sets, I suggest the authors analyzing these data, which can resolve this issue. Especially, I would like to know why *Hemibates stenosoma* shows a female-biased dimorphism in gill raker, despite it has a strong male-biased dimorphism in coloration.

4. Figure 2-4: Differences in gill raker length are strongly related to body size of the species and the extent of sexual size dimorphism. Considering these facts, I suggest that the authors should show relative differences in gill raker length of the species controlling their body size instead of actual and absolute differences in gill raker length.

5. Figure 2 and 4: If possible, please show the abbreviated scientific name of the cichlid fishes in the figure (like supplementary materials), so that the readers can easily collate the plot and species.

I hope these comments are useful to the authors.

Decision letter (RSPB-2019-0023.R0)

19-Feb-2019

Dear Professor Salzburger,

I am writing to inform you that we have now obtained responses from referees on manuscript RSPB-2019-0023 entitled "A functional trade-off between trophic adaptation and parental care leads to sexual dimorphism" which you submitted to Proceedings B.

I am sorry to have to tell you that, based on the advice of the Associate Editor and on the referees' reports, your manuscript has been rejected following full peer review. As the Associated Editor summarises below, the referees identified several major limitations of the manuscript: all of them commented on the lack of any phylogenetic component to the analysis, but several other concerns were also raised. Competition for space in Proceedings B is currently extremely severe, as many more manuscripts are submitted to us than we have space to print. We are therefore only able to publish those that are exceptional, convincing and present significant advances of broad interest, and must reject many good manuscripts.

Please find below the comments received from referees concerning your manuscript, not including confidential reports to the Editor. I hope you may find these useful should you wish to submit your manuscript elsewhere.

We are sorry that your manuscript has had an unfavourable outcome, but would like to thank you for offering your work to Proceedings B.

Yours sincerely,
Loeske Kruuk
Editor

Proceedings B
mailto: proceedingsb@royalsociety.org

Associate Editor
Board Member: 1
Comments to Author:

The referees all had several positive comments about the results, and their importance. Sexual dimorphisms are too often assumed to be due to sexual selection. Two of the referees raised concerns about the statistical analyses, however, and suggested that the authors need to incorporate a phylogenetic approach, as with any comparative study (or explain why this is unnecessary). One of the referees raised concerns about the novelty of the findings, and the origin of data, and wondered whether the results are based on reanalyses of previously published data. This issue should be clarified. Finally, I can understand why the authors rejected sexual selection but strictly speaking, this hypothesis would have to be tested before ruling it out (e.g., some cichlids use acoustic communication, and the mechanisms of sound production are not understood. It is far-fetched, but for all we know, these fish communicate using gill rakers).

Reviewer(s)' Comments to Author:

Referee: 1

Comments to the Author(s)

The manuscript "A functional trade-off between trophic adaptation and parental care leads to sexual dimorphism" provides an interesting analysis of gill raker sexual dimorphism in cichlid fishes from Lake Tanganyika. The hypotheses set up in this work are interesting. However, I have several strong reservations about the manuscript including the framing of the paper around the idea of optima, the novelty of data and breadth of the conclusions presented, and the rigor of the evolutionary analyses of trait associations.

The discussion of optima in the introduction is interesting and makes up a substantial part of the introduction and figure 1, but little about optima is really tested in the paper. There are now phylogenetic comparative methods available that would allow the testing of trait optima associated with particular ecological phenotypes such as:

For instance:

Mahler DL, Ingram T, Revell LJ, Losos JB. 2013 Exceptional convergence on the macroevolutionary landscape in island lizard radiations. *Science* 341, 292–295.

I felt the discussion of optima and the focus of figure 1 on optima and fitness could provide a misleading view of how these traits are diverging in these cichlid fishes and what the manuscript actually tests. The focus on optima in studies of adaptation has a long history of being criticized and the current discussion without any explicit tests of "optimality" I think is unjustified.

Also, it was unclear in several cases what was novel about the data presented. Were the values of stable isotopes taken from other published papers? If so, it would be good to clarify how much of this data is unique to this paper. Also, how many of the gill raker measurements are novel to this paper? It should be made clear how many are already published in Muschick et al. 2014 and what proportion/samples are novel to this work. Since the paper seems to largely center around the reanalysis of previously published data and the conclusions are not likely to have a large impact on our general understanding of sexual dimorphism, the general reuse of data to address the narrow questions posed about sexual dimorphism will likely have limited interest to a wide audience.

Finally, and perhaps most problematically, the data is not analyzed in a rigorous evolutionary framework. For instance, in Figure 2: Are the uni-parental male trait values used since these are predicted, as the manuscript makes a case, near the optima? For a correlation like this, I would assume only the male traits should be used. Also, if this correlation uses size-corrected gill raker lengths this should be stated in the figure axes and/or legend. Most critically, this correlation should be performed in a phylogenetically explicit framework. The current correlation is statistically invalid. Additionally, the current statistical analyses of sexual dimorphism in gill raker length does not seem to adequately account for phylogeny. A phylogenetic ANCOVA should be used to show that the gill raker lengths show differences in sexual dimorphism among the three different breeding modes. With only 20 species sampled, this is unfortunately unlikely to be significant for the different parental modes. However, without adequately accounting for phylogeny it is difficult to say that the correlations reported are not spuriously due to species sampling and an inflation of presumed power due to the OTUs being compared without a phylogenetic framework.

The manuscript also has a too substantial number of grammatical, formatting, or spelling mistakes.

Here are a few:

Line 29: "dimorphism in a primarily trophic structure"

Line 153: Strange font formatting "covered 65 species, well"

Line 560: "scull" should be "skull"

Referee: 2

Comments to the Author(s)

Simply put, this is one of the most fascinating and engaging manuscripts I have ever reviewed. I don't mean that the writing is great (it's fine), I mean the story is great. What a remarkable set of insights the authors have brought here. This is just glorious stuff. The idea that the gillrakers in uniparental species differ in length as a function of where the optimal length is for the feeding ecology is just outrageously interesting in the way that it produces so much diversity in raker length.

Now, having said that I am concerned about a serious methodological issue in the work, that may substantially weaken the results. The problem is that there are no phylogenetic corrections in any of the analyses, while the study is comparative in nature. In this day and age you just can't do this kind of a study without accounting for phylogenetic relationships among the taxa under consideration. I know it means a lot more work, and I also know it means the authors could well lose some of their significant results, but this needs to be dealt with. The relationships shown in Fig 2a, 3 and 4 all require the incorporation of phylogenetic relationships into any tests.

A second, but less concerning issue is that the way that trophic ecology is quantified (the C:N PC1) ultimately is unsatisfying when interpreting what is going on with the uniparental species. I think what one wants here is a more in depth discussion of how the gill rakers are functioning in these different species and what exactly the fish are eating. Is what they are eating strongly consistent with the story that is being told here that the magnitude and direction of dimorphism depends on the optimal length for the trophic habit? Hopefully there is more specific information available about what some of these fish are actually eating - beyond the hard-to-interpret isotope data.

It would also add to the story if the authors had some sense for how the gill rakers function in fish mouthbrooding eggs and larvae. What trophic features of gill rakers are problematic for brooding rakers? Why are rakers of different length adaptive for different prey? The issue here is that there should be a performance trade-off in the ecology, with long rakers being good for some prey but also for short rakers having a performance benefit for other prey. What are these performance benefits?

L149. Should be 'field' not 'filed'.

L168. It does not appear that any attempt was made to sample rakers from a specific region of the gill arch? Are rakers the same length at different positions along the arch? In many fish they are not. This would suggest that these relationships could be affected by where on the arch the rakers come from.

L210. Did you look to see if magnitude of dimorphism depends on body size? This could complicate analyses. I'm confused about whether this was accounted for here.

L223. Unclear what is meant by a 'scaled' PCA? I've never seen this term used before.

L276. Well, pelagic species that feed on tiny but mobile animals often have long gill rakers (as do sediment sifters that feed on tiny but mobile prey). But pelagic species that feed on large prey like fish do not have long gill rakers. None of these fish are really suspension feeders, they are suction feeders that are using gill rakers to prevent small, mobile prey from escaping from the buccal cavity into the opercular cavity after capture.

L283. I wish I could agree with this statement, but I think the authors are making assumptions about what prey underlie the isotope scores without showing the reader what they are actually eating. The manuscript shows shorter rakers in species that are low on the C:N axis and often longer rakers in fish that are high on the C:N axis (but definitely not always). But, if we don't actually know in this study what prey these fish are eating this creates some confusion.

-Peter Wainwright

Referee: 3

Comments to the Author(s)

RSPB-2019-0023: A functional trade-off between trophic adaptation and parental care leads to sexual dimorphism. By Ronco et al.

This is a very interesting study that investigated the relationship between sexual dimorphism in gill raker length and parental care patterns or trophic ecology. Using Tanganyika cichlid fishes with different parental care patterns and feeding ecology, they found that a functional trade-off between them, which may lead to differences in gill raker length between the sexes. The results of this study are clearly stated, which would add much contribution to our understanding of this study field. I have, however, a number of minor concerns, which were listed below.

1. Title: In general, sexual dimorphism means the dimorphism in secondary sex traits driven by sexual selection, such as coloration, weaponry and body size. However, this study deals with the sexual dimorphism in gill raker length. Hence, I suggest the authors adding some words in the title, e.g. "sexual dimorphism in gills of (cichlid) fishes".
2. Methods: The authors compared ecological traits among many cichlid species. Comparative methods, such as phylogenetic generalized least squares (PGLS), are generally used to examine the evolutionary process of these traits, but this study did not adopt such methods. Please clarify.
3. Discussion L314-330 and figure 3: The authors argued about sexual dimorphism within uni-parental mouthbrooders. I also agree that the differences in the extent of sexual dimorphism in gill rakers among uni-parental mouth brooders may be related to clutch size, egg size and the duration of parental care. If the authors have samples or data and find some papers describing these data sets, I suggest the authors analyzing these data, which can resolve this issue. Especially, I would like to know why *Hemibates stenosoma* shows a female-biased dimorphism in gill raker, despite it has a strong male-biased dimorphism in coloration.
4. Figure 2-4: Differences in gill raker length are strongly related to body size of the species and the extent of sexual size dimorphism. Considering these facts, I suggest that the authors should show relative differences in gill raker length of the species controlling their body size instead of actual and absolute differences in gill raker length.

5. Figure 2 and 4: If possible, please show the abbreviated scientific name of the cichlid fishes in the figure (like supplementary materials), so that the readers can easily collate the plot and species.

I hope these comments are useful to the authors.

Author's Response to Decision Letter for (RSPB-2019-0023.R0)

See Appendix A.

RSPB-2019-1050.R0

Review form: Reviewer 2

Recommendation

Accept as is

Scientific importance: Is the manuscript an original and important contribution to its field?

Excellent

General interest: Is the paper of sufficient general interest?

Excellent

Quality of the paper: Is the overall quality of the paper suitable?

Excellent

Is the length of the paper justified?

Yes

Should the paper be seen by a specialist statistical reviewer?

No

Do you have any concerns about statistical analyses in this paper? If so, please specify them explicitly in your report.

No

It is a condition of publication that authors make their supporting data, code and materials available - either as supplementary material or hosted in an external repository. Please rate, if applicable, the supporting data on the following criteria.

Is it accessible?

Yes

Is it clear?

Yes

Is it adequate?

Yes

Do you have any ethical concerns with this paper?

No

Comments to the Author

The authors have satisfied my previous concerns. I continue to maintain that this is an extremely interesting paper that provides an extremely novel explanation for the nature of sexual dimorphism.

Review form: Reviewer 4

Recommendation

Accept with minor revision (please list in comments)

Scientific importance: Is the manuscript an original and important contribution to its field?

Good

General interest: Is the paper of sufficient general interest?

Good

Quality of the paper: Is the overall quality of the paper suitable?

Good

Is the length of the paper justified?

Yes

Should the paper be seen by a specialist statistical reviewer?

No

Do you have any concerns about statistical analyses in this paper? If so, please specify them explicitly in your report.

No

It is a condition of publication that authors make their supporting data, code and materials available - either as supplementary material or hosted in an external repository. Please rate, if applicable, the supporting data on the following criteria.

Is it accessible?

Yes

Is it clear?

No

Is it adequate?

No

Do you have any ethical concerns with this paper?

No

Comments to the Author

RSPB-2019-1050

A functional trade-off between trophic adaptation and parental care predicts sexual dimorphism in cichlid fish

In this comparative study, the authors used 65 cichlid fishes from Lake Tanganyika, to explore whether gill rakers (the spine-like, bony protrusions of the gill arches of fishes) are sexually dimorphic and whether the degree of sexual dimorphism is related to breeding behaviour. Gill rakers are used primarily for food uptake and handling. Cichlid fishes either orally incubate their young for several weeks or guard them on the ground. Therefore, it stands to reason that mouthbrooding cichlids could have modified gill rakers compared to substrate guarding cichlids as modified rakers could serve to retain and manipulate eggs within the buccal cavity. Trophic ecology was estimated by stable isotope analyses on dried muscle tissue of nearly 700 individuals, with high nitrogen ($\delta^{15}\text{N}$) to carbon ($\delta^{13}\text{C}$) ratios indicating a high position in the foodweb (mainly pelagic cichlids) and low ratios indicating a lower trophic level (mostly benthic cichlid species). The researchers also measured gill rakers using a microscope and examined both sexes in around 20 of the 65 species. The researchers found sexual dimorphism in gill rakers but this was only apparent in the uniparental mouthbrooders. Biparental mouth brooders and fish that cared for their young on the ground did not show a noticeable difference in gill raker length. The paper is well written and clear but I have a few concerns that ought to be addressed before the paper will be ready for publication.

- 1) The authors don't explain sufficiently the link between sexual selection and parental care. This relationship typically dictates the extent of sexual dimorphism. Care usually suppress sexual selection. This link was missing (especially so in the abstract) and the dots need to be connected to make the logic more sound.
- 2) Why was it only the length of the gill rakers that were measured? Surely number and even width could also be important. Surely they too influence how the buccal cavity can hold eggs and larvae.
- 3) In what sense does this study help understand the causes of the evolution of sexual dimorphism? I would like to have a better understanding about what role diet plays in dictating gill raker length and then what % of the remaining variation does parental care mode explain. The framing for the study is that we will understand more about the evolution of sexual dimorphism because of this study...but I don't see this to be the case.
- 4) Do mouthbrooders have longer gill rakers than substrate guarders? I would like to know if the authors found such a relationship.
- 5) How did the isotope data map on to the diet items for these species?
- 6) L242. There must be great variation in what these animals eat depending on the location sampled, the time of year even the time of day sampled. In the ms, can you provide more of the details of how these issues were controlled for or how this variation was accounted for.

Minor Comments

L67. Respective empirical evidence- makes no sense in what sense is it respective? Respective to what?

L75. What is meant by trophic adaptation of a species

L77. Nest building and defense can both be sexually selected traits. Please reword.

L89. What is a trophic trait? Please define.

L92. Not clear why the breeding mode of the groups can predict the presence or absence of a conflicting function of that trait

L93. Gill raker morphology is not a setting.

L161. What part of the muscle was sampled? Was this data corrected for the mass of the muscle taken?

L 169. The number of males vs females measured for each species is reasonable for some species and tiny for others. Why were more samples not acquired?

L177. Surely the width and the number of gill rakers would also be super useful.

L175. Please provide a repeatability measure to show how close the two scorers were?

189. Shouldn't the size-corrected version be done on a model with all 65 species. Or should this correction be by food types or breeding mode?

L242. Many benthic species will be eating snails or pick invertebrates out of the water substrate boundary layer or will dig these out of sediment itself. I don't see how benthic animals would necessarily have a low trophic level.

Decision letter (RSPB-2019-1050.R0)

01-Jul-2019

Dear Professor Salzburger,

Your manuscript has now been peer reviewed and the reviews have been assessed by an Associate Editor. The reviewers' comments (not including confidential comments to the Editor) and the comments from the Associate Editor are included at the end of this email for your reference. As you will see, the new reviewer of the manuscript has raised some very valid comments and we would like to invite you to revise your manuscript to address these.

Research ethics:

Use of animals and field studies:

Please submit a copy of your revised paper within three weeks. If we do not hear from you within this time your manuscript will be rejected. If you are unable to meet this deadline please let us know as soon as possible, as we may be able to grant a short extension.

Best wishes,
Professor Loeske Kruuk
<mailto:proceedingsb@royalsociety.org>

Associate Editor Board Member

Comments to Author:

The manuscript has been improved, though the fourth reviewer points out several issues that should be addressed.

1. Please be sure to clarify the relationship between sexual selection and parental care.
2. Please explain why only the length of the gill rakers that were measured.
3. Please clarify how this study helps understand the evolution of sexual dimorphism in general. Also, please be sure to address the other issues raised by this reviewer. These will surely improve the paper further.

Reviewer(s)' Comments to Author:

Referee: 2

Comments to the Author(s).

The authors have satisfied my previous concerns. I continue to maintain that this is an extremely interesting paper that provides an extremely novel explanation for the nature of sexual dimorphism.

Referee: 4

Comments to the Author(s).

RSPB-2019-1050

A functional trade-off between trophic adaptation and parental care predicts sexual dimorphism in cichlid fish

In this comparative study, the authors used 65 cichlid fishes from Lake Tanganyika, to explore whether gill rakers (the spine-like, bony protrusions of the gill arches of fishes) are sexually dimorphic and whether the degree of sexual dimorphism is related to breeding behaviour. Gill rakers are used primarily for food uptake and handling. Cichlid fishes either orally incubate their young for several weeks or guard them on the ground. Therefore, it stands to reason that mouthbrooding cichlids could have modified gill rakers compared to substrate guarding cichlids as modified rakers could serve to retain and manipulate eggs within the buccal cavity. Trophic ecology was estimated by stable isotope analyses on dried muscle tissue of nearly 700 individuals, with high nitrogen ($\delta^{15}\text{N}$) to carbon ($\delta^{13}\text{C}$) ratios indicating a high position in the foodweb (mainly pelagic cichlids) and low ratios indicating a lower trophic level (mostly benthic cichlid species). The researchers also measured gill rakers using a microscope and examined both sexes in around 20 of the 65 species. The researchers found sexual dimorphism in gill rakers but this was only apparent in the uniparental mouthbrooders. Biparental mouth brooders and fish that cared for their young on the ground did not show a noticeable difference in gill raker length. The paper is well written and clear but I have a few concerns that ought to be addressed before the paper will be ready for publication.

- 1) The authors don't explain sufficiently the link between sexual selection and parental care. This relationship typically dictates the extent of sexual dimorphism. Care usually suppress sexual selection. This link was missing (especially so in the abstract) and the dots need to be connected to make the logic more sound.
- 2) Why was it only the length of the gill rakers that were measured? Surely number and even width could also be important. Surely they too influence how the buccal cavity can hold eggs and larvae.
- 3) In what sense does this study help understand the causes of the evolution of sexual dimorphism? I would like to have a better understanding about what role diet plays in dictating gill raker length and then what % of the remaining variation does parental care mode explain.

The framing for the study is that we will understand more about the evolution of sexual dimorphism because of this study...but I don't see this to be the case.

4) Do mouthbrooders have longer gill rakers than substrate guarders? I would like to know if the authors found such a relationship.

5) How did the isotope data map on to the diet items for these species?

6) L242. There must be great variation in what these animals eat depending on the location sampled, the time of year even the time of day sampled. In the ms, can you provide more of the details of how these issues were controlled for or how this variation was accounted for.

Minor Comments

L67. Respective empirical evidence- makes no sense in what sense is it respective? Respective to what?

L75. What is meant by tropic adaptation of a species

L77. Nest building and defense can both be sexually selected traits. Please reword.

L89. What is a trophic trait? Please define.

L92. Not clear why the breeding mode of the groups can predict the presence or absence of a conflicting function of that trait

L93. Gill raker morphology is not a setting.

L161. What part of the muscle was sampled? Was this data corrected for the mass of the muscle taken?

L 169. The number of males vs females measured for each species is reasonable for some species and tiny for others. Why were more samples not acquired?

L177. Surely the width and the number of gill rakers would also be super useful.

L175. Please provide a repeatability measure to show how close the two scorers were?

189. Shouldn't the size-corrected version be done on a model with all 65 species. Or should this correction be by food types or breeding mode?

L242. Many benthic species will be eating snails or pick inverts out of the water substrate boundary layer or will dig these out of sediment itself. I don't see how benthic animals would necessarily have a low trophic level.

Author's Response to Decision Letter for (RSPB-20191050.R0)

See Appendix B.

Decision letter (RSPB-2019-1050.R1)

30-Jul-2019

Dear Professor Salzburger

I am pleased to inform you that your manuscript entitled "A functional trade-off between trophic adaptation and parental care predicts sexual dimorphism in cichlid fish" has been accepted for publication in Proceedings B.

Open Access

Your article has been estimated as being 9 pages long. Our Production Office will be able to confirm the exact length at proof stage.

Paper charges

Sincerely,

Professor Loeske Kruuk
Editor, Proceedings B
<mailto:proceedingsb@royalsociety.org>

Associate Editor:

Board Member

Comments to Author:

The authors have addressed the reviewers' main concerns, and the manuscript has been improved.

Appendix A

Associate Editor Board Member: 1

Comments to Authors:

The referees all had several positive comments about the results, and their importance. Sexual dimorphisms are too often assumed to be due to sexual selection. Two of the referees raised concerns about the statistical analyses, however, and suggested that the authors need to incorporate a phylogenetic approach, as with any comparative study (or explain why this is unnecessary). One of the referees raised concerns about the novelty of the findings, and the origin of data, and wondered whether the results are based on reanalyses of previously published data. This issue should be clarified. Finally, I can understand why the authors rejected sexual selection but strictly speaking, this hypothesis would have to be tested before ruling it out (e.g., some cichlids use acoustic communication, and the mechanisms of sound production are not understood. It is far-fetched, but for all we know, these fish communicate using gill rakers)

RESPONSE: We are grateful to the overall positive response to our results and the comments on the importance of our work. As detailed below, we have now revised our manuscript to address the concerns of the referees with respect to the statistical analyses. Importantly, we now report the results from phylogenetic comparative analyses backing up our results (We would like to note here that the application of phylogenetic correction is per se not expected to have much of an effect in cases of adaptive radiations, where the different species are phylogenetic closely related and often equidistant from one another). Furthermore, we now clarify in the main text that our study is largely based on novel data (which was, in the previous version, mainly evident from the methods section and the supplementary material) and we emphasize the novelty of our results. Finally, we remain with our argumentation that sexual selection can be excluded as a driver for dimorphism in this case. We are not aware of any study linking sound production with the gill-raker apparatus in (cichlid) fish. Sound production has previously been linked to the pharyngeal jaw apparatus in cichlids (with a relevant study on this topic having been co-authored by the PI of the present study). However, while the pharyngeal jaw bones are developmentally related to gill-rakers, there is, to the best of our knowledge, no evidence of gill-rakers being involved in sound production in cichlids.

Reviewer(s)' Comments to Author:

Referee #1:

Comments to the Author(s)

The manuscript "A functional trade-off between trophic adaptation and parental care leads to sexual dimorphism" provides an interesting analysis of gill raker sexual dimorphism in cichlid fishes from Lake Tanganyika. The hypotheses set up in this work are interesting. However, I have several strong reservations about the manuscript including the framing of the paper around the idea of optima, the novelty of data and breadth of the conclusions presented, and the rigor of the evolutionary analyses of trait associations.

RESPONSE: We thank the referee for her/his general interest in our work and would like to respond to her/his major points below.

The discussion of optima in the introduction is interesting and makes up a substantial part of the introduction and figure 1, but little about optima is really tested in the paper.

There are now phylogenetic comparative methods available that would allow the testing of trait optima associated with particular ecological phenotypes such as: For instance: Mahler DL, Ingram T, Revell LJ, Losos JB. 2013 Exceptional convergence on the macroevolutionary landscape in island lizard radiations. *Science* 341, 292–295. I felt the discussion of optima and the focus of figure 1 on optima and fitness could provide a misleading view of how these traits are diverging in these cichlid fishes and what the manuscript actually tests. The focus on optima in studies of adaptation has a long history of being criticized and the current discussion without any explicit tests of "optimality" I think is unjustified.

REPNSE: The treatise on fitness optima is indeed an important part of the introduction, as this section builds up the hypothetical framework for our study. Although admittedly a simplification, the concept of fitness optima is central to the entire field of evolutionary biology and hence ubiquitous. In our case, we use this concept in a "theoretical point of view" (as specified at the beginning of this paragraph, L:66+), in particular to illustrate the resulting trade-off. In response to Reviewer #1, we have slightly modified Figure 1 and now depict the optima as discrete dashed lines (as done by many colleagues, see e.g. A.P. Hendry: *Eco-evolutionary Dynamics*). Reviewer #1 is correct in stating that we did not test much about optima in our paper. However, as clearly specified in our manuscript (see L:392-394), our aim is not to actually test trait optima or selection towards such an optimum. Instead, what we are interested in is the trade-off emerging from the existence of two different optima, a trophic and a reproductive one, leading to sexual dimorphism (see also the title of our manuscript).

The method suggested by Reviewer #1 would indeed be very interesting to investigate how trait evolution of gill raker morphology proceeded in the adaptive radiation and to pinpoint cases of convergent evolution across lineages. However, we feel that this would exceed the scope of this study, as the main focus of the manuscript is the observed sexual dimorphism in uni-parental mouthbrooders.

Also, it was unclear in several cases what was novel about the data presented. Were the values of stable isotopes taken from other published papers? If so, it would be good to clarify how much of this data is unique to this paper. Also, how many of the gill raker measurements are novel to this paper? It should be made clear how many are already published in Muschick et al. 2014 and what proportion/samples are novel to this work. Since the paper seems to largely center around the reanalysis of previously published data and the conclusions are not likely to have a large impact on our general understanding of sexual dimorphism, the general reuse of data to address the narrow questions posed about sexual dimorphism will likely have limited interest to a wide audience.

REPNSE: As specified in the methods section and the supplementary material of the original manuscript, the majority of the data presented in the manuscript has been newly acquired, including all stable isotope measurements and more than half of the gill raker length data. Importantly, most of the gill raker length data used to address the question of sexual dimorphism is new to this manuscript. The data from our previous study (Muschick et al. 2014) was basically used to establish the phenotype-environment correlation (Figure 2a and Supplementary Figure S1b) with all available data.

In order to make it more clear that most of the data are novel, we now state in the main text:

L148-154: “Combined with available data on gill raker length from additional Tanganyikan cichlid species [16], the final dataset covered 65 species, well representing the phylogenetic (13 out of 16 tribes [22]), eco-morphological, and behavioural (breeding modes) diversity of the species-flock of cichlid fishes in Lake Tanganyika (see electronic supplementary material for detailed information on sampling procedure and table S1 for sample sizes).”

And L176-178:

“We measured gill raker length in 508 specimens (38 species). In combination with data from Muschick et al. [16] we obtained a data set comprising 935 specimens and 65 species.”

This way, it should be clear that the majority of the data is novel even without consolidating “table S1 for sample sizes” as mentioned on L154.

We disagree with Reviewer #1 that our study is “not likely to have a large impact on our general understanding of sexual dimorphism”. Instead, we think that the finding of sexual dimorphism due to a functional trade-off between trophic adaptation and parental care is novel and should be of interest to a large audience working in this field. Our own assessment on this is supported by the other two referees. Reviewer #2 states, for example, that “this is one of the most fascinating and engaging manuscripts I have ever reviewed” and “The idea that the gill rakers in uniparental species differ in length as a function of where the optimal length is for the feeding ecology is just outrageously interesting”. Reviewer #3 finds that “The results of this study are clearly stated, which would add much contribution to our understanding of this study field”.

Finally, and perhaps most problematically, the data is not analyzed in a rigorous evolutionary framework. For instance, in Figure 2: Are the uni-parental male trait values used since these are predicted, as the manuscript makes a case, near the optima? For a correlation like this, I would assume only the male traits should be used.

RESPONSE: As mentioned above and below, we have analyzed our data in an evolutionary framework, which is now presented in the manuscript. These analyses lead to the same conclusions, which is not surprising in the case of an adaptive radiation where all species are closely related and many of them are phylogenetically equidistant (as they emerged from a recent common ancestor).

We do see the point of Reviewer#1 concerning the use of both sexes in the phenotype-environment correlation. However, the main goal of this analysis is to establish an across-taxon phenotype-environment correlation and how the observed diversity in gill raker length relates to the feeding ecology of the different species – and this should include both sexes. Further, the observed correlation is mainly influenced by substantial differences in gill raker length among the different species. The additional within species variance introduced by the sexual dimorphism does not change the conclusions of this analysis. When we exclude females and specimens without sex information of all uni-parental mouthbrooders, the results are comparable, but the taxon sampling is considerably reduced (54 instead of 65 species).

Also, if this correlation uses size-corrected gill raker lengths this should be stated in the figure axes and/or legend.

RESPONSE: We did use size corrected gill raker length, as described in detail in the Material and Methods section (d). For clarification we added this information to the figure axes and legend, which now reads: “Phenotype-environment correlation

between size-corrected gill raker length and trophic ecology (PC1-scores of stable isotope data)."

Most critically, this correlation should be performed in a phylogenetically explicit framework. The current correlation is statistically invalid. Additionally, the current statistical analyses of sexual dimorphism in gill raker length does not seem to adequately account for phylogeny. A phylogenetic ANCOVA should be used to show that the gill raker lengths show differences in sexual dimorphism among the three different breeding modes. With only 20 species sampled, this is unfortunately unlikely to be significant for the different parental modes. However, without adequately accounting for phylogeny it is difficult to say that the correlations reported are not spuriously due to species sampling and an inflation of presumed power due to the OTUs being compared without a phylogenetic framework.

RESPONSE: We agree with this and the other two referees that we should have presented our results in a phylogenetically explicit framework, and have extended the statistical analysis accordingly. To establish the phenotype-environment correlation we now additionally report results from a pGLS to account for phylogenetic relationship among the species. Further we used phylogenetic ANOVA to validate results of comparisons among breeding modes and with uniparental mouthbrooders. The conclusions remained largely unchanged.

The manuscript also has a too substantial number of grammatical, formatting, or spelling mistakes. Here are a few:

Line 29: "dimorphism in a primarily trophic structure"

RESPONSE: We are not quite sure what should be changed here.

Line 153: Strange font formatting "covered 65 species, well"

RESPONSE: Thank you for pointing this out, this was not obvious in the original word file.

Line 560: "scull" should be "skull"

RESPONSE: Thank you for pointing this typo out.

Referee #2:

Comments to the Author(s)

Simply put, this is one of the most fascinating and engaging manuscripts I have ever reviewed. I don't mean that the writing is great (it's fine), I mean the story is great. What a remarkable set of insights the authors have brought here. This is just glorious stuff. The idea that the gillrakers in uniparental species differ in length as a function of where the optimal length is for the feeding ecology is just outrageously interesting in the way that it produces so much diversity in raker length.

RESPONSE: We are more than honored to receive this feedback from such a well-known scientist.

Now, having said that I am concerned about a serious methodological issue in the work, that may substantially weaken the results. The problem is that there are no phylogenetic corrections in any of the analyses, while the study is comparative in nature. In this day and age you just can't do this kind of a study without accounting for

phylogenetic relationships among the taxa under consideration. I know it means a lot more work, and I also know it means the authors could well lose some of their significant results, but this needs to be dealt with. The relationships shown in Fig 2a, 3 and 4 all require the incorporation of phylogenetic relationships into any tests. **RESPONSE:** We agree with this critique (see also responses to Reviewers #1 and #3). In the revised version, we now present the results from all statistical analyses applying phylogenetic corrections. To establish the phenotype-environment correlation we now report results from a pGLS to account for phylogenetic relationship among the species. Further we used phylogenetic ANOVA to validate results of comparisons among breeding modes and with uniparental mouthbrooders. These analyses basically led to the same outcomes as before, so that our conclusions remain unchanged.

A second, but less concerning issue is that the way that trophic ecology is quantified (the C:N PC1) ultimately is unsatisfying when interpreting what is going on with the uniparental species. I think what one wants here is a more in depth discussion of how the gill rakers are functioning in these different species and what exactly the fish are eating. Is what they are eating strongly consistent with the story that is being told here that the magnitude and direction of dimorphism depends on the optimal length for the trophic habit? Hopefully there is more specific information available about what some of these fish are actually eating – beyond the hard-to-interpret isotope data.

RESPONSE: Prof. Wainwright is correct that stable isotope data alone are sometimes unsatisfying in the context of quantifying trophic ecology. On the other hand, stable isotope measurements are ideally suited to establish quantitative phenotype-environment correlations, for which categorical data (such as particular food types) are less well suited. In the case of the Tanganyikan cichlids, we have previously validated the use of stable isotope data by examining stomach contents in about 60 species (see e.g. Muschick et al. 2012). Other than that, the available information is largely observational. In response to Reviewer #2 we have added a new column to Table S1, in which we report the available food type information from observational data for the species tested for sexual dimorphism.

It would also add to the story if the authors had some sense for how the gill rakers function in fish mouthbrooding eggs and larvae. What trophic features of gill rakers are problematic for brooding rakers? Why are rakers of different length adaptive for different prey? The issue here is that there should be a performance trade-off in the ecology, with long rakers being good for some prey but also for short rakers having a performance benefit for other prey. What are these performance benefits?

RESPONSE: Unfortunately, very little is known about how gill raker morphology affects mouthbrooding performance. Regarding the trophic features, it has previously been established that planktonic feeders have longer rakers. In response to Reviewer #2 we have rephrased the corresponding section in the Discussion (see L:286+).

L149. Should be 'field' not 'filed'.

RESPONSE: Thank you for pointing this out.

L168. It does not appear that any attempt was made to sample rakers from a specific region of the gill arch? Are rakers the same length at different positions along the arch? In many fish they are not. This would suggest that these relationships could be affected by where on the arch the rakers come from.

RESPONSE: We did measure rakers taken from the same position on the gill arch. However, we did not explicitly state this in the previous version of the manuscript, but rather referred to a paper which describes the method in detail. We now clarified this by stating the following: L:172-175: *“Measurements were converted to millimetre scale and averaged across the three gill rakers measured per specimen (i.e., the second, third and fourth raker on the first branchial gill arch).”*

L210. Did you look to see if magnitude of dimorphism depends on body size? This could complicate analyses. I'm confused about whether this was accounted for here.

RESPONSE: We used size-corrected gill raker length for all analyses, as described in the Material and Methods section (d) and (e). The magnitude of sexual dimorphism of size independent gill raker length does not depend on body size of the species (this is based on a linear model: $R^2 = 0.0005$, $p = 0.92$).

L223. Unclear what is meant by a 'scaled' PCA? I've never seen this term used before.

RESPONSE: We used an ordinary PCA but we scaled the nitrogen and carbon isotope data before calculating the principal components to equally weight the two variables. We agree that this was not explicitly stated in the original version of the manuscript, and as scaling is commonly applied prior PCA, we simply removed the word *“scaled”*.

L276. Well, pelagic species that feed on tiny but mobile animals often have long gill rakers (as do sediment sifters that feed on tiny but mobile prey. But pelagic species that feed on large prey like fish do not have long gill rakers. None of these fish are really suspension feeders, they are suction feeders that are using gill rakers to prevent small, mobile prey from escaping from the buccal cavity into the opercular cavity after capture.

RESPONSE: Thank you very much for this feedback. We agree that we had oversimplified this relationship and therefore rephrased this section: L286-290: *“Gill rakers are an important structure for uptake and handling of food in the buccal cavity of fish [9], and the length of gill rakers is generally associated with different trophic ecologies: pelagic species feeding on small and mobile prey commonly have longer gill rakers, while benthic species feeding on larger or immobile prey have shorter gill rakers [10–16].”*

L283. I wish I could agree with this statement, but I think the authors are making assumptions about what prey underlie the isotope scores without showing the reader what they are actually eating. The manuscript shows shorter rakers in species that are low on the C:N axis and often longer rakers in fish that are high on the C:N axis (but definitely not always). But, if we don't actually know in this study what prey these fish are eating this creates some confusion.

RESPONSE: We agree, that this might be an over simplification. We therefore rephrased and extended this section to justify our interpretation. Additionally, we discuss an example of a mismatch in this relationship:

L293-304: *“As expected, we find an association between gill raker length and trophic ecology (as approximated by the PC1 of stable isotope data), with longer gill rakers in cichlids with more pelagic stable isotope signatures, and shorter gill rakers in species with more benthic signatures (figure 2a). Based on a previous study linking stable isotope signatures with stomach content analysis in Tanganyika cichlids [22], we can conclude that pelagic stable isotope signatures usually correspond to invertebrate/zooplankton/small fish feeders, whereas species with benthic signatures*

predominantly feed on algae and plants. Note, however, that also predatory species feeding on large fish show pelagic signatures, but have rather short gill rakers (see e.g. Lepidolamprologus profundicula; 'Leppo'; figure 2a)."

-Peter Wainwright

Referee #3:

Comments to the Author(s)

RSPB-2019-0023: A functional trade-off between trophic adaptation and parental care leads to sexual dimorphism. By Ronco et al.

This is a very interesting study that investigated the relationship between sexual dimorphism in gill raker length and parental care patterns or trophic ecology. Using Tanganyika cichlid fishes with different parental care patterns and feeding ecology, they found that a functional trade-off between them, which may lead to differences in gill raker length between the sexes. The results of this study are clearly stated, which would add much contribution to our understanding of this study field. I have, however, a number of minor concerns, which were listed below.

RESPONSE: We thank Referee #3 for his overall positive feedback on our work and the valuable suggestions to improve it further.

1. Title: In general, sexual dimorphism means the dimorphism in secondary sex traits driven by sexual selection, such as coloration, weaponry and body size. However, this study deals with the sexual dimorphism in gill raker length. Hence, I suggest the authors adding some words in the title, e.g. "sexual dimorphism in gills of (cichlid) fishes".

RESPONSE: In sexual selection literature, the term sexual dimorphism does not imply per se that it is due to sexual selection, which is why we would like to keep the title as is. However, we slightly modified the title in the revised version to include the "cichlid fish" notion.

2. Methods: The authors compared ecological traits among many cichlid species. Comparative methods, such as phylogenetic generalized least squares (PGLS), are generally used to examine the evolutionary process of these traits, but this study did not adopt such methods. Please clarify.

RESPONSE: We agree on this and therefore revised the manuscript to now also report comparative methods (see also response to Referees #1 and #2). To establish the phenotype-environment correlation we now report results from a pGLS to account for phylogenetic relationship among the species. Further we used phylogenetic ANOVA to validate results of comparisons among breeding modes and with uniparental mouthbrooders. These analyses basically led to the same outcomes as before, so that our conclusions remain largely unchanged.

3. Discussion L314-330 and figure 3: The authors argued about sexual dimorphism within uni-parental mouthbrooders. I also agree that the differences in the extent of sexual dimorphism in gill rakers among uni-parental mouth brooders may be related to clutch size, egg size and the duration of parental care. If the authors have samples or data and find some papers describing these data sets, I suggest the authors analyzing these data, which can resolve this issue. Especially, I would like to know

why *Hemibates stenosoma* shows a female-biased dimorphism in gill raker, despite it has a strong male-biased dimorphism in coloration.

RESPONSE: Unfortunately, we do not have and did not find enough (reliable) data on such life history traits for the species we tested for sexual dimorphism. However, in response to this referee, we added the following sentence to the discussion and added the supplementary figure S2a:

L351-355: “*Unfortunately, data on life history traits is too scarce (and/or too vague) to allow testing for an association with gill raker length. Nevertheless, clutch size emerges as a promising candidate trait to explain variation in the direction of sexual dimorphism among uni-parental mouthbrooders (see electronic supplementary material, figure S2a).*”

Concerning *Hemibates stenosoma* (and also other species), we do not think (and the data shows no evidence) that the observed sexual dimorphism correlates with the degree of sexual colour dimorphism of the species.

4. Figure 2-4: Differences in gill raker length are strongly related to body size of the species and the extent of sexual size dimorphism. Considering these facts, I suggest that the authors should show relative differences in gill raker length of the species controlling their body size instead of actual and absolute differences in gill raker length.

RESPONSE: We size-corrected gill raker length prior analysis, as described in the Material and Methods section (d). To state this more prominently, we added this information to all figure legends and repeatedly mention it in the main text.

5. Figure 2 and 4: If possible, please show the abbreviated scientific name of the cichlid fishes in the figure (like supplementary materials), so that the readers can easily collate the plot and species.

RESPONSE: We thank reviewer #3 for pointing this out, we adjusted the two figures accordingly.

I hope these comments are useful to the authors.

Appendix B

Reviewer(s)' Comments to Author:

Referee: 2

Comments to the Author(s).

The authors have satisfied my previous concerns. I continue to maintain that this is an extremely interesting paper that provides an extremely novel explanation for the nature of sexual dimorphism.

RESPONSE: We thank Referee #2 for his enthusiastic feedback on our study!

Referee: 4

Comments to the Author(s).

RSPB-2019-1050

A functional trade-off between trophic adaptation and parental care predicts sexual dimorphism in cichlid fish

In this comparative study, the authors used 65 cichlid fishes from Lake Tanganyika, to explore whether gill rakers (the spine-like, bony protrusions of the gill arches of fishes) are sexually dimorphic and whether the degree of sexual dimorphism is related to breeding behaviour. Gill rakers are used primarily for food uptake and handling. Cichlid fishes either orally incubate their young for several weeks or guard them on the ground. Therefore, it stands to reason that mouthbrooding cichlids could have modified gill rakers compared to substrate guarding cichlids as modified rakers could serve to retain and manipulate eggs within the buccal cavity. Trophic ecology was estimated by stable isotope analyses on dried muscle tissue of nearly 700 individuals, with high nitrogen ($\delta^{15}\text{N}$) to carbon ($\delta^{13}\text{C}$) ratios indicating a high position in the foodweb (mainly pelagic cichlids) and low ratios indicating a lower trophic level (mostly benthic cichlid species). The researchers also measured gill rakers using a microscope and examined both sexes in around 20 of the 65 species. The researchers found sexual dimorphism in gill rakers but this was only apparent in the uniparental mouthbrooders. Biparental mouth brooders and fish that cared for their young on the ground did not show a noticeable difference in gill raker length. The paper is well written and clear but I have a few concerns that ought to be addressed before the paper will be ready for publication.

RESPONSE: We thank Referee #4 for the overall very positive feedback on our work and her/his suggestions to further improve the manuscript. A detailed response to the open questions can be found below.

1) The authors don't explain sufficiently the link between sexual selection and parental care. This relationship typically dictates the extent of sexual dimorphism. Care usually suppress sexual selection. This link was missing (especially so in the abstract) and the dots need to be connected to make the logic more sound.

RESPONSE: We agree with Referee #4 that there can be a link between sexual selection and parental care. However, in our case, the link between parental care and sexual selection is of minor importance because (i) all cichlid species (also non-mouthbrooders) provide intensive parental care (mentioned for Lake Tanganyika cichlids on L101-105); and (ii) in uniparental species, sexual selection could potentially lead to sexual dimorphism due to unequal parental investment, however a

trophic trait such as gill rakers, is highly unlikely to be target of a sexual selection (as discussed in L417-L424). To emphasise this and to make the link between sexual selection, parental care and our case, we reworded the abstract: L26-29: *“Here we report a case where a sex-specific conflicting functional demand related to parental care but not to sexual selection explains sexual dimorphism in a primarily trophic structure, the gill rakers of cichlid fishes.”*

2) Why was it only the length of the gill rakers that were measured? Surely number and even width could also be important. Surely they too influence how the buccal cavity can hold eggs and larvae.

RESPONSE: The reason why we focused on gill raker length in this study is that gill raker length shows a strong association with trophic ecology in fish, including in cichlids, which is now clearly spelled out in the Introduction section (L96-100). Further gill raker length is the very component of the gill raker apparatus that varies the most between recently diverged fish. For example, it has previously been shown in stickleback fish that gill raker length is more variable than gill raker number (Berner et al. 2010, Mol. Ecol.), and in a Tanganyika cichlid, we found that different populations differ in gill raker length but not in number (Theis et al. 2014, Mol. Ecol.). We agree with Referee #4 that it would be interesting to examine other components of the trophic morphology of cichlids in the context of sexual dimorphism and have included this notion as an outlook sentence in the Discussion section (L439-441).

3) In what sense does this study help understand the causes of the evolution of sexual dimorphism? I would like to have a better understanding about what role diet plays in dictating gill raker length and then what % of the remaining variation does parental care mode explain. The framing for the study is that we will understand more about the evolution of sexual dimorphism because of this study...but I don't see this to be the case.

RESPONSE: Our study presents a so far unrecognized scenario of how sexual dimorphism may evolve WITHOUT the action of sexual selection or initial niche divergence between sexes (or, in the words of Referee #2: "provides an extremely novel explanation for the nature of sexual dimorphism"). [Note that the evolutionary causes of sexual dimorphism are usually attributed to sexual selection (Andersson, 1994; Darwin, 1871) or initial niche divergence (Slatkin, 1984; Temeles et al., 2000).] In order to emphasize this novelty aspect of our study, we have included the notion “we present a so far unrecognised scenario for the evolution of sexual dimorphism” in the Abstract (L38-41) and mention it in the Discussion section (L426-429). Thus, our study not only contributes to the understanding of the evolution of sexual dimorphism, but also opens a new perspective on how a functional trade-off can act as an additional source of morphological variation in a trophic trait.

Regarding the role of diet, we would like to refer to the results presented in Figure 2, where we show that diet plays a major role in predicting gill raker length. This matches findings in other fish species, as reported on L96-100. Such a strong phenotype-environment correlation is one of the key features of an adaptive radiation (see Schluter 2000) and, at the same time, provides strong evidence for the adaptive nature of the trait in question.

It would indeed be interesting to be able to directly test whether the breeding mode of a species causes a deviation from the gill raker length predicted from the trophic specialisation. However, in our opinion, our data does not provide the necessary power to do so: First, while stable isotope measurements provide a solid proxy for the overall ecology of a species, we feel that the so obtained indirect measurements of trophic ecology are too vague to build a model. Second, the model would need to be fitted with nest guarders only, and as the non-mouthbrooders are monophyletic in Lake Tanganyika (with one exception, *Boulengerochromis microlepis*, which is not included in our study), the model would have a strong phylogenetic bias.

4) Do mouthbrooders have longer gill rakers than substrate guarders? I would like to know if the authors found such a relationship.

RESPONSE: As evident from Figure 2a, overall gill raker length does not differ among the breeding modes, with mouthbrooders spanning the whole spectrum of gill raker lengths observed in Lake Tanganyika cichlids.

5) How did the isotope data map on to the diet items for these species?

RESPONSE: A detailed relationship between stomach contents and stable isotope measurements has previously been established for Lake Tanganyika cichlids (see e.g. Muschick et al. 2012, CurrBio; Muschick et al. 2014, ProcB). In particular, in the study of Muschick et al. 2012, the close connection between food items and nitrogen stable isotope data becomes evident for the set of species investigated in the present manuscript (see Figure 2 in Muschick et al. 2012; please also note the reference signatures from potential food items such as algae, plants, crabs, mussels, etc.). The comparison between food items and stable isotope data in Muschick et al. 2012 is mentioned in detail in L161-163 and L299-304 of the present manuscript and additional information on feeding ecology is provided in the supplementary table S1 (see also response to point 6).

6) L242. There must be great variation in what these animals eat depending on the location sampled, the time of year even the time of day sampled. In the ms, can you provide more of the details of how these issues were controlled for or how this variation was accounted for.

RESPONSE: In fact, most cichlid species are highly specialized to a particular food type (see also Figure 2 in Muschick et al. 2012, panels B, D and E), and there is usually very little variation in what these fishes eat between sample localities or seasons. The exceptions are a handful of generalist species, which can be identified by larger confidence intervals in Figure 2 in Muschick et al. 2012 (but even the generalists do not nearly span the entire food spectrum). Importantly, stable isotope measurements from muscle tissue integrate food-uptake over a period of ~6 months in tropical fish and thus control for potential occasional variation. Overall, this makes stable isotopes more robust to sampling stochasticity (time, location) than other assessments of trophic ecology, such as through stomach contents. Finally, we note that Lake Tanganyika is a tropical lake and therefore underlies relatively little seasonal fluctuations. For all these reasons, we are convinced that stable isotope signatures provide an adequate – and arguably the best available – metric to quantify trophic ecology of Lake Tanganyika cichlids.

Minor Comments

L67. Respective empirical evidence- makes no sense in what sense is it respective?
Respective to what?

RESPONSE: We thank Referee #4 for pointing this out. We have rephrased this sentence accordingly. It now reads (L66-67): "*Yet, empirical evidence for ecological sex traits remains scarce [7,8].*"

L75. What is meant by trophic adaptation of a species

RESPONSE: The notion "trophic trait" was indeed not defined in the manuscript. We have now included a definition of the term and have changed the sentence to:

L74-76: "*For example, a structure involved in food uptake and/or processing (i.e. a trophic trait) of a species could have an additional function (...)*"

L77. Nest building and defense can both be sexually selected traits. Please reword.

RESPONSE: We do see the point of Referee #4 that there is a possibility that nest building, if occurring before mating, could be under sexual selection. We have rephrased this section to:

L74-77: "*For example, a structure involved in food uptake and/or processing (i.e. a trophic trait) of a species could have an additional function in a reproductive behaviour without sexual selection acting on the focal trait, such as in nest-building or defending offspring [1].*"

L89. What is a trophic trait? Please define.

RESPONSE: In the new version, we define "trophic trait" at its first occurrence at L75 (see response above).

L92. Not clear why the breeding mode of the groups can predict the presence or absence of a conflicting function of that trait

RESPONSE: We thank Referee #4 for pointing out that this section was a bit unclear. We have now rephrased it to:

L89-96: "*The gill rakers of cichlid fishes from East African Lake Tanganyika provide a rare opportunity to test in a comparative framework for a sex-specific trade-off related to brood care – but not to sexual selection – in an otherwise trophic trait. This is because of the important role of gill rakers (i.e. spine-like, bony protrusions of the branchial gill arches in fishes) in food uptake and handling of particles within the buccal cavity [9]; the potential involvement of gill rakers in brood care in many cichlids; and the different brood care strategies found among the closely related cichlids from Lake Tanganyika.*"

L93. Gill raker morphology is not a setting.

RESPONSE: We thank Referee #4 for pointing this out, we have rephrased the entire section (see above).

L161. What part of the muscle was sampled? Was this data corrected for the mass of the muscle taken?

RESPONSE: We sampled muscle tissue from the epaxialis between the head and the dorsal fin of each specimen, which is now mentioned in the manuscript (L164-165). We would like to note that we previously verified, in a pilot experiment (Master Thesis of Anna Boila, unpublished), that the location of the biopsy has no effect on the measurement. Concerning the correction for mass: As mentioned in the methods section (L159) we used stable isotope ratios, i.e. the ratio between the rare isotope over the common isotope, which is independent of the mass sampled.

L 169. The number of males vs females measured for each species is reasonable for some species and tiny for others. Why were more samples not acquired?

RESPONSE: The main reasons for the difference in sample size among the species is that not all cichlid species occur at equal frequencies in Lake Tanganyika, and that some are much more difficult to sample than others. Importantly, however, we have large samples sizes in the focal groups (mouthbrooders).

L177. Surely the width and the number of gill rakers would also be super useful.

RESPONSE: We agree with Referee #4 that additional measurements of other components of trophic morphology would be interesting. As mentioned above, we focused on gill raker length, because this is the main component of gill rakers showing ample and ecology-dependent variation in cichlids and other fish species. However, we now also mention that other components should be studied in the future (L439-441).

L175. Please provide a repeatability measure to show how close the two scorers were?

RESPONSE: As described in the Methods section of the manuscript (L178-179) the specimens were randomly assigned to the two investigators. Moreover, gill rakers were measured directly from the gill apparatus that was removed from the specimens prior to measuring. Measuring thus occurred blinded with respect to the species and sex of the specimens (L174-175). With this strategy, we can exclude the possibility that trait differences among species and/or the sexes within species are due to potential investigator-biases.

189. Shouldn't the size-corrected version be done on a model with all 65 species. Or should this correction be by food types or breeding mode?

RESPONSE: We agree with Referee #4 that the size correction could be done in several ways and if direct comparisons between species are made, size correction should be done using a common (65 species) linear model (as we did for the overall phenotype-environment correlation). Importantly, however, the analysis of sexual dimorphism focused on within species comparisons (i.e., males versus females). In this case, a species-specific size correction is preferred because it accounts for species-specific differences in the size-dependence of the trait.

L242. Many benthic species will be eating snails or pick inverts out of the water substrate boundary layer or will dig these out of sediment itself. I don't see how benthic animals would necessarily have a low trophic level.

RESPONSE: We agree with Referee #4 that particular benthic species may not necessarily be lower in the food chain compared to some of the pelagic species.

However, our data show that lower PC1-scores of stable isotope values occurred in benthic/littoral species, which – in case of Tanganyika species – are predominantly algae grazers. To make this distinction clear, we added the term 'littoral' to the respective text passage:

L241-245: *“Higher PC1-scores thus reflected pelagic feeding (e.g., on zooplankton and/or fish fry) and a relatively high position in the food chain (hereafter simply referred to as 'pelagic'), whereas benthic/littoral species with a mainly algivorous feeding lifestyle and a lower trophic position had lower PC1-scores (hereafter simply called 'benthic').”*